# Identification of BoLA Alleles Associated with BLV Proviral Load in US Beef Cows

**DOI:** 10.3390/pathogens11101093

**Published:** 2022-09-24

**Authors:** Ciarra H. LaHuis, Oscar J. Benitez, Casey J. Droscha, Sukhdeep Singh, Andrew Borgman, Chaelynne E. Lohr, Paul C. Bartlett, Tasia M. Taxis

**Affiliations:** 1Department of Animal Science, College of Agriculture and Natural Resources, Michigan State University, East Lansing, MI 48824, USA; 2School of Veterinary Medicine, Texas Tech University, Lubbock, TX 79415, USA; 3CentralStar Cooperative, Lansing, MI 48910, USA; 4Department of Plant, Soil and Microbial Sciences, College of Agriculture and Natural Resources, Michigan State University, East Lansing, MI 48824, USA; 5Borgman Consulting Group LLC, Alma, MI 48801, USA; 6Large Animal Clinical Sciences, College of Veterinary Medicine, Michigan State University, East Lansing, MI 488424, USA

**Keywords:** beef cattle, BLV, BoLA DRB3, bovine leukemia virus, disease progression, disease resistance

## Abstract

Bovine leukemia virus (BLV) causes enzootic bovine leukosis, the most common neoplastic disease in cattle. Previous work estimates that 78% of US beef operations and 38% of US beef cattle are seropositive for BLV. Infection by BLV in a herd is an economic concern for producers as evidence suggests that it causes an increase in cost and a subsequent decrease in profit to producers. Studies investigating BLV in dairy cattle have noted disease resistance or susceptibility, measured by a proviral load (PVL) associated with specific alleles of the bovine leukocyte antigen (BoLA) DRB3 gene. This study aims to investigate the associations between BoLA DRB3 alleles and BLV PVL in beef cattle. Samples were collected from 157 Midwest beef cows. BoLA DRB3 alleles were identified and compared with BLV PVL. One BoLA DRB3 allele, **026:01*, was found to be associated with high PVL in relation to the average of the sampled population. In contrast, two alleles, **033:01* and **002:01*, were found to be associated with low PVL. This study provides evidence of a relationship between BoLA DRB3 alleles and BLV PVL in US beef cows.

## 1. Introduction

Bovine leukemia virus (BLV) is a delta retrovirus and the etiological agent causing enzootic bovine leukosis in cattle. Approximately 89% of dairy and 78% of beef operations in the US have at least one BLV-infected animal in the herd [1,2]. Additionally, 38% of US beef cattle and 29% of Midwest beef cattle were found to be seropositive for BLV [2,3]. The transmission of BLV may occur with the reuse of hypodermic needles, direct contact, dehorning tools, examination sleeves, or by blood-sucking insects [4,5,6]. Neighboring animals within an infected herd pose a significant risk of BLV transmission [7]. One infected animal can lead to multiple infected animals within the herd. 

There is a range of clinical signs of BLV infection. Between 60 and 70% of infected animals remain aleukemic, having normal lymphocyte counts [8]. Approximately 30% of infected animals progress to persistent lymphocytosis, characterized by an increased risk of infection by opportunistic pathogens [9]. A small percentage (2–5%) of infected animals develop lymphoma, leading to the condemnation at slaughter of both dairy and beef animals [10,11]. Malignant lymphoma accounts for 22% of the cause for condemnation at slaughter for beef and dairy cattle in the Great Lakes region of the US and 13.5% for beef cattle in the US and is a direct profit loss to producers [10,11]. A quantitative polymerase chain reaction (qPCR) assay can be used to determine the concentration of the BLV provirus in a blood sample, associating the proviral load (PVL) with the stage of disease, where animals with a greater PVL are indicative of a more severe infection and a potentially increased risk of transmission of the provirus infectious agent to their herdmates [12,13]. 

Host genetics may play a role in BLV disease progression. The major histocompatibility complex (MHC) is composed of genes involved in antigen presentation to T cells [14]. In cattle, the MHC gene region is termed the bovine leukocyte antigen (BoLA). In cattle, the MHC Class II *BoLA-DRB3* gene locus is highly polymorphic, with an identified 384 alleles [15]. Multiple studies have linked variations in the *BoLA-DRB3* gene locus to levels of PVL in dairy cattle [16,17]. The role of *BoLA-DRB3* alleles in BLV disease progression in beef cattle is largely unknown. The current study aims to identify the potential associations between *BoLA-DRB3* alleles and BLV disease progression in a population of beef cows from the Midwest region of the US. 

## 2. Results and Discussion

After enrolling cows with a known BLV antibody presence, a qPCR test revealed that PVLs in the sampled beef population ranged from 0.00 to 2.54 BLV copies/Bos β-actin copies, with a mean equal to 0.52 and a median of 0.24 (Appendix A). The animals with undetectable PVL were included in the analysis because a PVL of zero with a positive BLV ELISA result may indicate disease resilience by -*BoLA-DRB3* alleles.

Lymphocyte counts (LC) were observed as an average per allele, though no trend was identified. This is likely due to the limited dataset. Previous publications have identified a correlation between BLV PVL and LC in addition to the observed association between BLV PVL and *DRB3* alleles [16,17,18]. Therefore, it is likely that there may be an association between *DRB3* allele and LC. Future research may aim to identify the potential association between *DRB3* allele and LC. 

Alleles **009:02*, **010:01*, **011:01* have been associated with resistance to BLV disease progression in infected dairy cows. In contrast, alleles **012:01* and **015:01* have been associated with susceptibility to BLV disease progression, potentially leading to persistent lymphocytosis or lymphoma [16,17]. Four out of these five alleles were also identified in the sampled beef population (Table 1). 

Similarly to what has been identified in dairy cattle, allele **002:01* was associated with low PVL in the sampled population of Midwest beef cows [16,18]. The animals with allele **002:01* were found to have approximately one-third of the PVL in comparison to the average of the sampled population (Table 1). Additionally, the animals with allele **033:01* were found to have a PVL of less than one-tenth of the sampled population average (Table 1). To date, no publications have associated allele **033:01* with BLV PVL in beef or dairy cattle. 

Allele **026:01* has been reported at a frequency of between 1 and 3% in populations of Baggara, Kenana, and Butana cattle [19]. However, in the current study, allele **026:01* was present at a higher frequency (20.36%), and animals with the allele were found to have a BLV PVL approximately 3 times greater than the average of the sampled population (Table 1). Allele **026:01* may potentially associate with a greater susceptibility for BLV disease progression in beef cattle. A lower population frequency of **026:01* may indicate a decreased likelihood for disease progression in BLV-infected beef herds. 

In the present study, 18 of the known 384 *BoLA-DRB3* alleles were identified (Table 1). Of the 18 alleles, 9 were noted in Simmental cattle from Columbia, but publications regarding *BoLA-DRB3* allele frequencies within US Angus and Simmental cattle are nonexistent [20]. The relationship between BLV PVL and *BoLA-DRB3* alleles can be observed in Figure 1, where the estimated allelic effect is the deviation from the average PVL at the allele from the average PVL of the sampled population (0.52 BLV copies/Bos β-actin copies). The publications observing *BoLA-DRB3* alleles in dairy cattle have found a similar number of alleles in populations approximately doubled in size [16]. The greater allelic diversity observed in beef cows may be a result of the differences in effective population size between the beef and dairy industries [21,22]. The allelic diversity in the study population could be increased further with a larger population of animals from various regions outside the Midwest US. Additionally, the sampled population is limited to Angus and Simmental breeds. Greater diversity in beef breeds may also increase allelic diversity.

Upon evaluating the effect of *BoLA-DRB3* genotypes on PVL in the study population, 3 out of 33 genotypes were significant (Appendix A). *BoLA-DRB3 *026:01/*026:01* was associated with a PVL of 0.74 times, or nearly three-quarters, that of the population average. Genotypes *BoLA-DRB3 *026:01/*002:01* and *BoLA-DRB3 *018:01/*018:01* were associated with PVLs of 0.67 and 0.63 times that of the population average. The study population size is a limitation. The effect of *BoLA-DRB3* genotypes on PVL should be evaluated with a larger, more diverse population sample. 

## 3. Materials and Methods

### 3.1. Samples

All animals were approved for use by the Institutional Animal Care and Use Committee. Blood samples were collected from Angus, Simmental, and Angus x Simmental crossed beef cows aged 24–168 months (n = 157) from 9 Michigan and Iowa beef cow–calf operations (Appendix A) [2]. Immediately following blood sample collection, LC was assessed as previously described [23]. Cows with a known presence of BLV antibodies, tested by enzyme-linked immunosorbent assay (ELISA) were selected. Whole blood was collected by coccygeal venipuncture from each selected cow and stored at −80 °C until DNA extraction. 

### 3.2. Animals PVL Quantification

DNA extraction was performed using the DNeasy Blood and Tissue Kit (Qiagen, Hilden, Germany). DNA quantity and quality was determined using the NanoDrop One/One^c^ (ThermoFisher Scientific, Austin, TX, USA). The methods to determine PVL followed Pavliscak et al., 2020 [24], and PVL was reported as a ratio of BLV polymerase gene copies to Beta Actin gene copies. 

### 3.3. BoLA-DRB3 Allele Determination

The *BoLA-DRB3* exon 2 was amplified from each DNA sample. Following Lohr et al., 2022 [16], two master mixes were prepared with separate tagged primers specific to exon 2 of the *BoLA-DRB3* gene (Table 2). Separate tagged primers allowed for multiplex sequencing by combining the following in a master mix: 25 µL 2X DreamTaq PCR Master Mix (ThermoFisher Scientific, Austin, TX, USA); 0.5 µL DRB3.1F or DRB3.4F forward primer; 0.5 µL DRB3.R reverse primer; 20.5 µL water; and 3.5 µL DNA for each reaction. All reactions were performed using Applied Biosystems 2720 thermal cycler 96 well (ThermoFisher Scientific, Austin, TX, USA) with the following conditions: 95 °C for 2 min, 34X (95 °C for 30 s, 68 °C for 30 s, 72 °C for 30 s), then 72 °C for 10 min. Amplicon size was confirmed by running a 1.5% agarose gel at 110 V for 50 min. 

Following confirmation of amplicon size for each sample, the DNA was sequenced by Illumina MiSeq. The *BoLA-DRB3* allele determination followed that of Lohr et al., 2022 [16], except for the heterozygous genotypes, which required at least 29% of the reads to align to the called reference allele. The homozygous genotypes required at least 72% of the reads to align to the called reference allele. 

### 3.4. Statistical Analysis

Statistical analysis was performed using SAS 9.4 (SAS Institute Inc 2013, Cary, NC, USA). The proviral load was log transformed to stabilize the variance and minimize the skewness of the residuals. The statistical model used to analyze the data was:
$$y_{\mathrm{ij}}=\mu +\beta ({Ử}_{i}-\overset{-}{x})+{\mathrm{bola}}_{\mathrm{j1}}+{\mathrm{bola}}_{\mathrm{j2}}+e_{i}$$
where y is the response (log PVL) for the *i*th cow having the *BoLA-DRB3* genotype j = [j_1_j_2_]; β is the regression coefficient on cow age x_i_, expressed as the deviation from the mean cow age $\overset{-}{x}$; bola_j1_ and bola_j2_ are the random effects of the 2 alleles j_1_ and j_2_ at the *BoLA-DRB3* gene locus; e_i_ is the environmental effect (or measurement error) related to the observation on the *i*th cow. The allelic effects at the *BoLA-DRB3* gene locus having allelic variance component σ^2^_bola_ were modeled as normally, independently, and identically distributed random additive effects within each cow. The combined variance due to the *BoLA-DRB3* gene locus was 2σ^2^_bola_. The statistical model is similar to that shown in Saama et al., 2004 [25], which had the random effect of the BoLA *DRB3.2* locus. Treating allelic effects as random is useful when there are some alleles with low frequencies relative to the other alleles in the population [26]. 

The effect of the genotype on PVL was analyzed using proc glimmix in SAS 9.4 (SAS Institute Inc 2013, Cary, NC, USA) with the genotype as the fixed effect in the model. The data met the normal distribution assumption. The genotypes with 9 or more observations were used in this analysis. The post hoc mean comparison was performed using Tukey’s adjustment with a significance level of 0.05.

## 4. Conclusions

Novel associations were found between the *BoLA-DRB3* alleles and BLV PVL in the sampled population of US Midwest beef cows. Further research is needed to include a larger, more diverse population. Additionally, obtaining one time point for measurement of PVL does not provide a measure of the disease’s endemic steady state or disease progression. Therefore, it may be valuable to longitudinally measure PVL in a BLV-infected beef cow population and determine the *BoLA-DRB3* alleles to achieve a measure of disease progression. With more evidence, the beef industry may consider selecting cattle for breeding that have resistance to BLV disease progression and that are less infectious to their herdmates as measured by PVL. 

## Figures and Tables

**Figure 1 pathogens-11-01093-f001:**
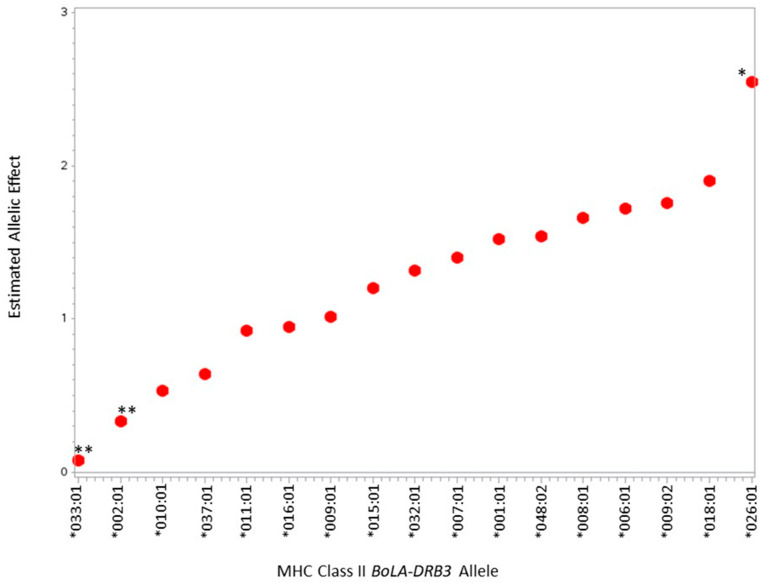
Estimated effect of MHC Class II *BoLA*-*DRB3* alleles on BLV proviral load in beef cattle. Red dots indicate the *BOLA-DRB3* allele estimated allelic effect as a deviation from the population average PVL. Asterisks represent significance, with * 0.05 ≤ *p* ≤ 0.10 and ** *p* ≤ 0.05.

**Table 1 pathogens-11-01093-t001:** Estimated allele frequencies and association between *BoLA-DRB3* alleles and bovine leukemia virus (BLV) proviral load (PVL) in beef cattle.

Allele	Total Count ^1^	# of Animals ^2^	Allele Frequency	Estimated Allelic Effect ^3^	*p*–Value ^4^	Lymphocyte Count (#/μL) ^5^
*010:01	1	1	0.003	0.53	0.60	3934 ± 0
*001:01	5	5	0.016	1.52	0.68	6957.80 ± 2369.64
*011:01	1	1	0.003	0.93	0.95	6823 ± 0
*015:01	3	2	0.010	1.20	0.86	6033 ± 782
*016:01	4	3	0.013	0.95	0.96	4789.50 ± 625.97
*018:01	99	74	0.315	1.90	0.21	8494.57 ± 379.70
*002:01	92	66	0.293	0.33	0.04 **	5628.72 ± 258.81
*026:01	66	46	0.210	2.55	0.08 *	8394.68 ± 434.96
*032:01	6	6	0.019	1.31	0.79	8193.17 ± 1651.94
*033:01	10	9	0.032	0.08	0.01 **	6531 ± 1236.41
*037:01	1	1	0.003	0.64	0.71	6602 ± 0
*048:02	4	2	0.013	1.54	0.65	5832.50 ± 405.59
*006:01	2	1	0.006	1.72	0.61	14475 ± 0
*007:01	7	4	0.022	1.40	0.68	5670.71 ± 336.23
*008:01	7	5	0.002	1.66	0.56	7296.43 ± 995.11
*009:01	3	2	0.010	1.01	0.99	7044 ± 1618
*009:02	2	1	0.006	1.76	0.60	9641 ± 0

^1^ Number of times the allele was identified in the US Midwest beef cow population. ^2^ Number (#) of animals harboring each allele. ^3^ Estimated allelic effects are shown as deviations from the average PVL in the population. A value of 1 indicates that the allelic effect at the respective allele is equal to the population average PVL. ^4^ ** *p* ≤ 0.05, * 0.05 ≤ *p* ≤ 0.10. ^5^ Lymphocyte count is shown with standard error.

**Table 2 pathogens-11-01093-t002:** BoLA-DRB3 exon 2 primers.

Primer	Direction	Sequence ^1^	Length (bp) ^2^	Tm ^3^
*DRB3.1F*	Forward	ACACTGACGACATGGTTCTACA **TCGTGGAGCG** ATCCTCTCTCGCAGCACATTTCC	55	70.5
*DRB3.4F*	Forward	ACACTGACGACATGGTTCTACA **TGCCTGGTGG** ATCCTCTCTCGCAGCACATTTCC	55	70.8
*DRB3.R*	Reverse	TACGGTAGCAGAGACTTGGTCT TCGCCGCTGCACAGTGAAACTCTC	46	70

^1^ Bold text of primer sequence highlights the unique barcode allowing two animals to be sequenced within a well. ^2^ Number of base pairs included in the primer. ^3^ Temperature at which the primer is optimal for PCR.

## Data Availability

Not applicable.

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
