# Peer review of "Identification of BoLA Alleles Associated with BLV Proviral Load in US Beef Cows"

_pathogens, 2022, doi:10.3390/pathogens11101093_

Round 1
Reviewer 1 Report
The study conducted by LaHuis et al. titled ' identification of BoLA Alleles Associated with BLV Disease 2 Progression in US Beef Cows" is interesting. I have only concern that is about the data which is currently available in a very limited form. Moreover, the alleles in gene associated with Enzootic Bovine Leukosis are still not proper validation
Reviewer 2 Report
Comments to authors:
The manuscript by LaHuis and colleagues describes the association between the level of BLV proviral load (PVL) and BoLA DRB3 alleles in a population of US beef cattle. The association of BoLA alleles with BLV proviral load may be of interest in particular cases where PVL has been shown to be indicator of resistance to BLV or indicator of progression of infection. However, the manuscript has some shortcomings, concerning the study population (size, composition, breeds), statistical analysis and presentation of results.
Major comments:
- Title: BLV disease progression is not a direct consequence of proviral load. This term should be replaced in title. Also, the breeds included in the sampled population should be mentioned in title.
- Summary, lines 22/23: the authors have only measured proviral load (PVL) not resistance or susceptibility.
- Data on the proviral load, which is the main dependent variable measured in this study is not presented. The description of the BLV phenotypes found in the study population should be shown as to appreciate the level, variability and distribution of the PVL in the sampled population.
- It should also be noted that the determination of PVL in a unique sample may not be representative of a BLV steady state. This is one limitation of this study and should be clearly explained.
- Statistical Analysis (line 133): the statistical model does not consider the interaction of the two alleles, please explain.
- Lines 140 and 141: regarding the variance due to BoLA gene locus, the cited reference is not appropriate, as it refers to another measured variable (response to dexamethasone) dependent on BoLA genotype, not BLV PVL. The variance due to BoLA locus, and also the variance due to environmental effect (error) are results of the study, and should be given in the Results section.
- Table 1: The frequency of some alleles seems to be relatively high. Is the sampled population representative of the study population? Are the BoLA allele frequencies obtained comparable to those obtained from US beef cattle, or from same breed beef cattle anywhere?
Table 1: Please include a column indicating the number of animals harboring each allele (as to see the degree of homocygosis)
Table 1: Please explain how “estimated allelic effect” should be interpreted by readers.
- More data on the study population should be given in Material and Methods. For the association to be valid, it is important to see if alleles came from different sires.
- Figure 1 is redundant, as the data on allelic effect is numerically presented on Table 1, this figure does not contribute additional information to the paper.
- Due to the considerations given above, limitations of the study should be clearly addressed (sample size, breed, composition of sample, PVL measured in a unique sample, etc).
- Conclusion: the dependent measured variable is BLV PVL, which is not synonymous with BLV disease progression. Lines 148 to 150 is not founded on results, no inferences about disease progression or infectiousness can be done from results.
Minor comments:
- Some references are not adequate or wrongly cited throughout the text (Example: reference 15 in line 54; reference 9 in line 43; reference 12 in line 53). Main references on association of BoLA alleles with BLV proviral load should be given in line 54.
- Page 1, line 41. Aleukemic should replace aleukemia
- line 56: BLV disease progression was not evaluated in the present study
- Lines 72-73: Is the lack of diversity found within expected values, considering that the sample came from 9 different herds? The authors claim that this is due to the limited size of population and to limited breeds. Please explain. This fact also limits the obtained results.
-Lines 74/75: The statement on sampled breeds corresponds to Material and Methods section, not results
- line 126: please check typing of genotype.
- Lines 143/144: Data from the animal with the allele *027:03 should be excluded from Table 1 if not included in the analysis.
Reviewer 3 Report
Review of the manuscript no.: pathogens-1798198
The manuscript by LaHuis et al. describes the approach which aims to investigate which specific BoLA-DRB3 alleles are associated with progression of BLV infection. In my opinion the topic of the manuscript in general is interesting but description of the subject in very condensed. It is worth to be published, however there are some points which should be addressed by the authors to improve the clarity of the text.
Major points:
In general the paragraph Results and Discussion is very condensed and fits rather to the Communication not the full article according to Pathogens standards. I assume this is due to the fact that the approach was very simple but maybe there are some more information in the results which could be presented and discussed as well.
1. Authors do not mention whether there were any animals with undetectable PLV in the tested group. Considering the aim I assume that not, but there is no clear statement. There is also no information about the distribution of PVLs in the tested population only the average value.
2. In this paragraph authors reveal that tested cows belonged to two breeds, but such information is missing in Material and Methods.
Additionally, I agree that greater diversity in breed may increase allelic diversity (line 75), but since there are already these two breeds, is there any association between alleles distribution or PVLs and a breed?
3. I think it would be worth to mention whether the animals with the discussed alleles were homo or heterozygotes.
4. There is no information about ELISA test used for serodiagnostics but in line 143 authors mention that allele *027:03 was removed from statistical analysis due to missing ELISA and PVL values. So, why the animal was considered in the whole analysis if its serological and PVL status was unknown.
5. Since the main aim of the study is to find the alleles associated with enzootic bovine leukosis progression it would be worth to measure the leukocyte counts in the animals with different alleles. It could be interesting information, additional to PVL counts.
Minor points:
Line 100-101: The information about the method used for serodiagnostics is missing.
Line 110: The . BoLA-DRB3 exon 2 was rather amplified not isolated. I would suggest the change of the verb.
Reviewer 4 Report
I recommend publishing this article as short communication
Introduction
Authors should discuss the economic importance of the disease
Results
Line 68, do you have not symbol on p value in table 1 and you did not clarified the meaning of each p value
Discussion
The results were not interpreted well with the previous literatures
Round 2
Reviewer 2 Report
After revising the second version of the manuscript, I see the main observations I have made in my first revision, were not fully answered.
In this report I transcribe my original observations, the authors’ answer (in blue) and explain why they were not fully answered (in italics)
1. Data on the proviral load, which is the main dependent variable measured in this study is not presented. The description of the BLV phenotypes found in the study population should be shown as to appreciate the level, variability and distribution of the PVL in the sampled population.
The PVL range, mean, and median are now included in Results and Discussion (Lines 64-68).
The PVL range, mean and median values are insufficient for the readers to see the behaviour of the variable in the study population. You should give at least the distribution of PVL by allele, genotype and breed. Individual data of each animal (genotype and PVL) should also be presented as a supplementary table.
2. It should also be noted that the determination of PVL in a unique sample may not be representative of a BLV steady state. This is one limitation of this study and should be clearly explained.
Thank you for your comment, we have recognized this limitation (Lines 196-197).
The sentence included in the Conclusions section of the revised manuscript does not clearly explain the limitation I have observed.
3. - Statistical Analysis (line 133): the statistical model does not consider the interaction of the two alleles, please explain.
We agree that genotypes are important to consider, and we plan to address this in future research with a larger dataset. Given the low genotypic distribution in our dataset, we have chosen to exclude this information from the manuscript.
For clarity to the readers, the reason why genotypes were not included in the analysis (low genotypic distribution of the dataset) should be clearly stated in the manuscript. As this is an important issue of the study population, it should also be resumed in the discussion.
4. Table 1: The frequency of some alleles seems to be relatively high. Is the sampled population representative of the study population? Are the BoLA allele frequencies obtained comparable to those obtained from US beef cattle, or from same breed beef cattle anywhere?
Publications regarding DRB3 allele frequencies within US Angus and Simmental beef cattle are non-existent. See publication below regarding frequencies in Simmental cattle (n=60) from Columbia. Nine of the alleles noted in the Columbian Simmental population were also found in our sampled US population, however, in differing frequencies. Many of the alleles Ordonez et al. (2022) noted were not found in our sampled population.
Ordonez, D., Bohorquez, M.D., Avendano, C., Patarroyo, M. A. (2022) Comparing class II MHC DRB3 diversity in Columbian simmental and simbrah cattle across worldwide bovine populations. Frontiers in Genetics, 13.
Given that no information is available on BoLA DRB3 diversity in US beef cattle, it would be desirable to compare allele frequencies with Angus and Simmental cattle from other locations (for example with those from Ordenez et al and others).
5. Table 1: Please include a column indicating the number of animals harboring each allele (as to see the degree of homocygosis).
We do find importance in taking genotypes into consideration. Given the low distrubution of genotypes in our dataset, we have chosen to exclude this information from the manuscript.
This is a key point that, in my opinion, should not be excluded from the dataset.
6. Table 1: Please explain how “estimated allelic effect” should be interpreted by readers.
“Estimated allelic effects are shown as deviations from the average PVL in the population.” (Line 84)
The explanation given by authors is insufficient. In which case the estimated allelic effect is positive (that is, it tends to increse the PVL) or negative ?
7. More data on the study population should be given in Material and Methods. For the association to be valid, it is important to see if alleles came from different sires.
We agree. Currently, this is out of the scope of this paper and the dataset. Future research aims to address this comment.
Again, this point, which is crucial for the correct interpretation of the associations, even more given the unusual high frequencies observed in the study population (3 alleles accounting for more than 80% of the allele frequency) was not considered by authors.
8. Figure 1 is redundant, as the data on allelic effect is numerically presented on Table 1, this figure does not contribute additional information to the paper.
While redundant information, we feel a visual representation of the data can be valuable for interpretation
Figure 1 was maintained in the revised manuscript. Instead, a figure showing variability of PVL by allele and breed should be more illustrative for readers.
Minor comments:
The authors did not answered speciffically to each comment, but responded to them globally.
- Some references are not adequate or wrongly cited throughout the text (Example: reference 15 in line 54; reference 9 in line 43; reference 12 in line 53). Main references on association of BoLA alleles with BLV proviral load should be given in line 54.
This point has not been absolutely considered by authors.
- Lines 72-73: Is the lack of diversity found within expected values, considering that the sample came from 9 different herds? The authors claim that this is due to the limited size of population and to limited breeds. Please explain. This fact also limits the obtained results.
The sentence added in lines 105-106 of the revised manuscript is not appropriate to explain the lack of diversity found, as the cited study was carried out on a « phenotipically selected population for genotyping » (page 5 of the cited paper).
- Lines 143/144: Data from the animal with the allele *027:03 should be excluded from Table 1 if not included in the analysis.
We retained information about allele *027:03 to note the presence in the sampled beef cattle. However, the allele was removed from downstream analysis because of the missing metadata. If you’d prefer, we are happy to remove the allele from our results and only mention it was found in the sampled population.
Again, I insist on excluding data from the animal with the allele *027:03 from the table, if it was not included in the analysis.
Reviewer 3 Report
Review of the manuscript no.: pathogens-1798198
I have read the corrected version of the manuscript no. pathogens-1798198 by LaHuis et al. and in my opinion the additional information and corrections included by the author improved the quality of the text. However I still have one concern about the description of the data. Authors added the information about the lymphocyte counts in the tested animals (lines 66-67 Table 1) but did not include any description in Material and Methods paragraph. I think it should be corrected before publication.
